# The red blood cell distribution width is associated with all-cause and cardiovascular mortality among individuals with non-alcoholic fatty liver disease

Yingxiu Huang🆔, Ting Ao, Yinying Wang, Peng Zhen, Ming Hu🆔*

Department of Infectious Disease, Beijing Luhe Hospital, Capital Medical University, Beijing, China

* hmyx2012@sina.com

## Abstract

### Background

Identifying reliable prognostic indicators is essential for the appropriate management of non-alcoholic fatty liver disease (NAFLD).
Red blood cell distribution width (RDW) has been established as an inflammatory marker associated with cardiovascular outcomes. This study aimed to evaluate the association between RDW and both cardiovascular and all-cause mortality in individuals with NAFLD.

### Methods

Data from 7,438 participants with NAFLD were analyzed, collected between 2005 and 2016 through the National Health and Nutrition Examination Survey (NHANES). Mortality data were retrieved from the National Death Index (NDI). Restricted cubic spline (RCS) analysis was used to illustrate the relationship between RDW and mortality risk, Weighted Cox proportional hazards models were used to assess the independent relationship between RDW and mortality risk. Receiver operating characteristic (ROC) curves were generated to evaluate the predictive ability of RDW for survival outcomes.

### Results

During a median follow-up period of 124 months, 1,269 deaths were recorded, including 335 from cardiovascular causes. RDW positively correlated with both cardiovascular and all-cause mortality according to the RCS analysis. Participants were categorized into quartiles based on RDW levels. Those in the highest RDW quartile (Q4) demonstrated a significantly higher risk of cardiovascular mortality (HR 3.61, 95% confidence interval [CI]:2.17–6.02, P=0.009) and all-cause mortality (HR 2.29, 95% CI:1.72–3.06, P < 0.0001), according to the weighted Cox hazards models. Additionally, the area under the curve (AUC) for all-cause mortality at 3, 5 and 10 years was, 0.69, 0.67, and 0.66, respectively. For cardiovascular mortality, the AUCs were 0.70, 0.68, and 0.68, respectively.

**Data availability statement:** All original data could be publicly available at the NHANES database: https://wwwn.cdc.gov/nchs/nhanes/Default.aspx. The data supporting the conclusions of this article have been uploaded to the Zenodo database (https://doi.org/10.5281/zenodo.14862447), without undue reservation.

**Funding:** The author(s) received no specific funding for this work.

**Competing interests:** The authors have declared that no competing interests exist.

## Conclusion

Among patients with NAFLD, RDW was identified as an independent predictor of increased cardiovascular and all-cause mortality risk.

## Introduction

Non-alcoholic fatty liver disease (NAFLD), a widespread metabolic disorder, is characterized by substantial hepatic fat accumulation, systemic inflammation, and insulin resistance [1]. NAFLD is the most prevalent chronic liver disease affecting up to 40% of the general population [1,2]. It progresses through stages of steatosis, fibrosis/cirrhosis, and steatohepatitis, potentially leading to liver failure or hepatocellular cancer, both of which have poor prognosis and low survival rates [3,4]. Extensive research has consistently demonstrated a strong association between cardiovascular disease (CVD) and NAFLD [2]. Independent of traditional cardiovascular risk factors, NAFLD is closely linked to an increased risk of severe cardiovascular events and comorbidities [3]. NAFLD is strongly associated with diabetes, chronic renal disease, and cardiovascular disease [5], and is especially frequent in individuals with hepatocellular carcinoma [6], posing a considerable potential health and economic cost to society. To improve the monitoring of NAFLD and its associated mortality, identifying an affordable and readily available prognostic metric is crucial.

Red blood cell distribution width (RDW), a routinely measured parameter in a complete blood count (CBC), assesses the variation in the size of red blood cells (RBCs). RDW is calculated by determining the standard deviation of RBC volume relative to mean corpuscular volume (MCV). The normal range for RDW is typically 11–15% [7]. While low RDW values are generally not clinically significant, elevated RDW levels can be associated with increased levels of inflammatory markers such as erythrocyte sedimentation rate (ESR), C-reactive protein (CRP), and interleukins (IL) [7]. Numerous studies have demonstrated that RDW can serve as a prognostic marker for various diseases, including acute myocardial infarction[8], COVID-19[9], heart failure [10], sepsis [11], and hepatocellular carcinoma [12]. Although several studies have explored the association between RDW and various conditions, fewer have focused on its relationship with NAFLD, Yang *et al.* [13]reported that patients with NAFLD tended to exhibit elevated RDW levels in a Chinese hospital cohort. However, in the United State, the relationship between RDW and mortality in patients with NAFLD remains unclear.

To address this gap, this study used a comprehensive population-based survey to evaluate the association among RDW, all-cause mortality, and cardiovascular mortality in individuals with NAFLD, thereby providing valuable insights into the health status of the US population.

## Materials and methods

### Study design and participants

This study used data from the NHANES database, which is conducted by the National Center for Health Statistics (NCHS) of the Centers for Disease Control and Prevention (CDC)[14]. Information on the general nutritional condition and state of health of Americans who were not in institutions was gathered using nationally representative NHANES survey [15]. All participants provided informed consent, and the NHANES dataset does not contain any individually identifiable patient data. The NHANES protocols were approved by the National Center for Health Statistics' Institutional Review Board. This study included data collected between 2005 and 2016. The study sample comprised adults aged 20 years and older with available

data to determine NAFLD status (n = 5,872). Participants were excluded if they had evidence of secondary causes of liver disease, such as excessive alcohol intake[16] (≥4 drinks per day for men and ≥3 drinks per day for women; one drink is defined as a 12 ounce beer, a 5 ounce glass of wine, or 1.5 ounces of liquor, including whiskey, gin, beer, wine, wine coolers, and any other alcoholic beverage, viral hepatitis B or C, or were pregnant (Fig 1).

## RDW measurement

RDW (percentage) was measured using a Coulter analyzer at mobile examination centers (MEC), using peripheral blood samples [17].

## NAFLD measurement

A Fatty Liver Index (FLI) of ≥ 60 indicated the existence of NAFLD, provided that the following conditions were absent: (1) hepatitis B infection (indicated by a positive hepatitis B surface antigen) or hepatitis C infection (evidenced by a positive hepatitis C antibody or HCV RNA); (2) excessive alcohol consumption (for women, three alcoholic drinks per day, and for men, more than four). The FLI was calculated using the following formulae [18]:

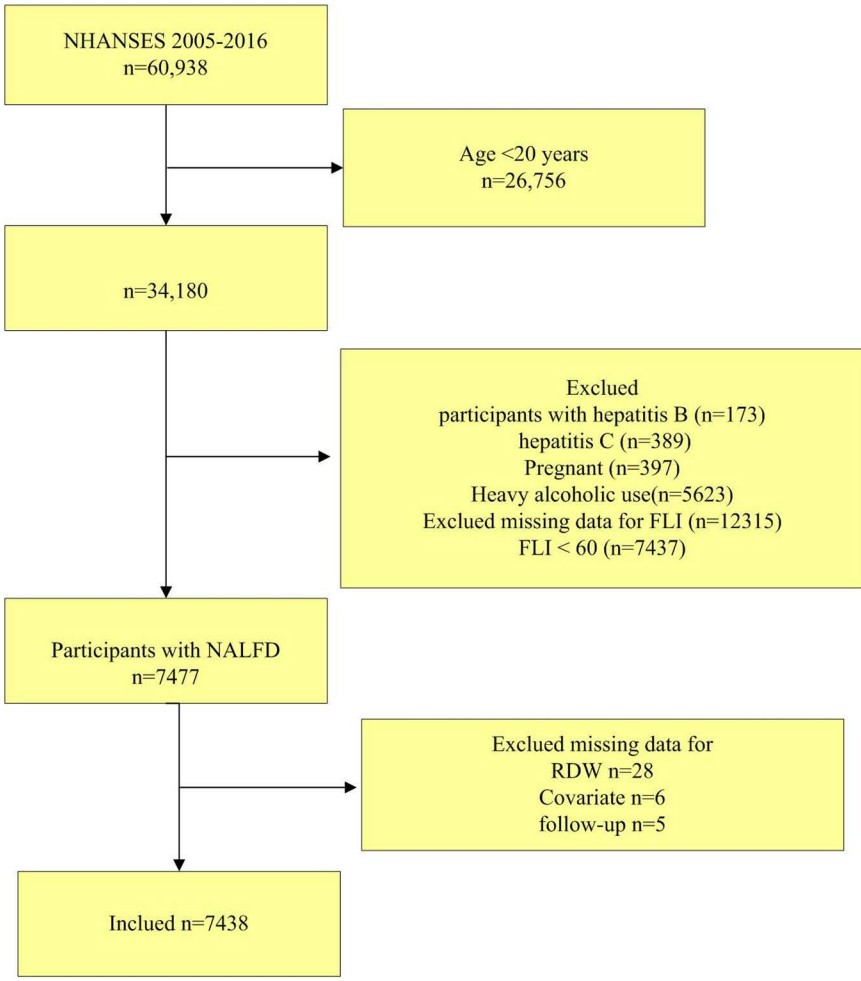

**Fig 1. Flowchart of study.**

$$\text{FLI} = \left( e^{0.953 \times \ln(\text{triglycerides, mg/dL}) + 0.139 \times (\text{BMI, kg/m2}) + 0.718 \times \ln(\text{GGT, U/L}) + 0.053 \times (\text{waist circumference, cm}) - 15.745} \right) / $$

$$\left( 1 + e^{0.953 \times \ln(\text{triglycerides, mg/dL}) + 0.139 \times (\text{BMI, kg/m2}) + 0.718 \times \ln(\text{GGT, U/L}) + 0.053 \times (\text{waist circumference, cm}) - 15.745} \right) \times 100$$

## Evaluation of mortality from all causes and follow-up

Mortality status was determined by integrating information obtained from the National Death Index, which may be reached at https://www.cdc.gov/nchs/data-linkage/mortality-public.htm, with the NHANES data. Participants were classified as either alive or deceased based on NDI data. The follow-up period was calculated by subtracting the date of the NHANES examination from the date of death (December 31, 2019). Cardiovascular and all-cause mortality were retrieved and analyzed using the 10th Revision of the International Classification of Diseases (ICD-10). Cardiovascular mortality was defined as death due to cardiac conditions, which were categorized under the codes I00–I09, I11, I13, and I20–I51. The median follow-up period was 124 months (interquartile range: 100–147).

## Covariates

Potential confounding factors were considered based on previous research and clinical judgment, including age, gender, educational level, marital status, body mass index (BMI), smoking status, CVD, diabetes, alcohol intake, and hypertension. Age was treated as a continuous factor, while gender was divided into male and female categories. Race/ethnicity was categorized as Mexican American, non-Hispanic Blacks, non-Hispanic Whites, and other. Marital status was classified as married or living with a partner, or living alone. Educational level was classified into three groups: less than 9 years, 9–12 years, and more than 12 years. Smoking status was classified as never, former, or current smoker. Alcohol use was categorized as never, former or current [14]. Diabetes was defined using a comprehensive approach that included a hemoglobin A1c level of 6.5%, a fasting blood glucose level of 126 mg/dL, use of oral hypoglycemic agents or insulin, or a self-reported history of diabetes [19]. Hypertension was defined as systolic blood pressure ≥140 mmHg or diastolic blood pressure ≥90 mmHg, or by a self-reported history of hypertension or the use of oral antihypertensive medications [19]. Self-reported information regarding cardiovascular disease (CVD) history included prior diagnoses of heart failure, coronary heart disease, angina, heart attack, or stroke [14].

## Statistical analysis

This analysis accounts for the complex NHANES sample design by incorporating appropriate sample weights, stratifications, and clustering. Sample weights were calculated by dividing the 2-year MEC weight by six. Categorical variables were presented as proportions (%), while continuous variables were presented as mean (standard deviation, SD) or median (interquartile range, IQR), as appropriate. Categorical variables were compared using the survey-weighted chi-squared test, whereas continuous variables were compared using survey-weighted linear regression.

Baseline characteristics were grouped according to RDW quartiles (Q1 (≤12.2), Q2 (12.3–12.7), Q3 (12.8–13.4), and Q4 (≥13.5)). To investigate possible nonlinear correlations between all-cause and cardiovascular mortality among NAFLD patients, RCS with four knots (5th, 35th, 65th, and 95th percentiles) were employed.

Survey-weighted Cox proportional hazards models were used to assess the independent association between RDW and all-cause and cardiovascular mortality in patients with NAFLD. The results were displayed across three models: Model 1, unadjusted; Model 2, which

was modified for race, sex, age, marriage; and Model 3, additionally adjusted for BMI and smoking, diabetes, hypertension, and history of CVD.

Kaplan–Meier survival analysis employed the log-rank test to examine the odds of survival for individuals with NAFLD who were categorized by RDW quartile group. Variables including smoking status, sex, BMI (< 30 and ≥ 30 kg/m2), diabetes, age (< 65 and ≥ 65 years), and CVD history, were the basis for the stratified and interaction analyses.

A time-dependent receiver operating characteristic (ROC) curve was used to evaluate how well RDW predicted survival outcomes at various time intervals.

Data analysis was conducted using R software version 4.2.2, R survey package version 4.2.2, and Free Statistics software version 1.9.2 [20]. Statistical significance was asset at a two-tailed P value < 0.05. This cohort study was conducted according to the guidelines outlined in the Strengthening the Reporting of Observational Studies in Epidemiology (STROBE) statement [21].

## Results

### Participants demographics at baseline

The final analysis included 7,438 participants with NAFLD, representing a weighted sample size of 48,125,092 Americans age ≥ 20 years (Table 1). The mean age of the participants was 51.85 years (SD: 15.27 years), and 53% of them were men. Most of the participants were non-Hispanic white (71.24%).

### Relationships between RDW and all-cause mortality in NAFLD

During a median follow-up period of 124 months, 1,269 deaths occurred, including 335 cardiovascular deaths. A positive nonlinear correlation between RDW and all-cause mortality was shown by RCS analysis (P for nonlinear < 0.001) (Fig 2A).

In Model 1, higher RDW values were associated with a substantial increase in the risk of all-cause death (HR 1.30, 95% confidence interval [CI] 1.24–1.36, p < 0.001) (Table 2). After multivariate correction, every unit increase in RDW was linked to a 24% increase in mortality risk in Model 2 (HR 1.24, 95% CI 1.16–1.33, p < 0.001) and a 22% increase in risk in Model 3 (HR 1.22, 95% CI 1.14–1.31, p < 0.001) (Table 2). When analyzed as a categorical variable in Model 3, individuals in the highest RDW Q4 group had a markedly elevated risk of all-cause mortality (HR 2.29, 95% CI 172–3.06, p < 0.001) was higher than that of the Q1 group (table 2).

According to the survival curve analysis, those in the higher RDW Q4 group had a significantly lower survival rate than those in the lower RDW group (P<0.001) (Fig 3A).

Subgroup examining the association between RDW and all-cause and mortality across age, sex, BMI, diabetes, smoking status, and history of CVD revealed no significant interactions (P for interaction > 0.05) (Fig 4A).

### Relationships between cardiovascular death and RDW in individuals with NAFLD

In Model 1, a notable increase in cardiovascular mortality was observed with increasing RDW (HR 1.33, 95% CI 1.27–1.39, p < 0.001) (Table 2). After extensive adjustments, each unit increased in RDW corresponded to a 23% increase in cardiovascular mortality risk (Model 3, HR 1.23, 95% CI 1.14–1.33, p < 0.001) (see Table 2). When analyzed as a categorical variable, individuals with the highest RDW Q4 group had a notably higher risk of cardiovascular mortality (HR 3.61, 95% CI 2.17–6.02, p <0.001) compared to Q1 group in model 3 (Table 2).

**Table 1. Characteristics of participants by quartile of the RDW.**

| Characteristics | Total (n = 7438) | RDW(%) | | | | P value |
|---|---|---|---|---|---|---|
| | | Q1 (≤12.2) (n = 1580) | Q2 (12.3–12.7) (n= 2040) | Q3 (12.8–13.4) (n =1846) | Q4 (≥13.5) (n =1972) | |
| Weighted number | 48125092 | 12193717 | 14582568 | 11112543 | 10236265 | |
| Age, years | 54.2 ± 16.1 | 48.8 ±15.7 | 53.3 ±15.6 | 56.1 ±16.0 | 57.6 ±15.9 | < 0.001 |
| Sex, male, n (%) | 3862 (51.9) | 923 (58.4) | 1142 (56) | 966 (52.3) | 831 (42.1) | < 0.001 |
| Race, n (%) | | | | | | < 0.001 |
| Mexican American | 1263 (17.0) | 334 (21.1) | 389 (19.1) | 306 (16.6) | 234 (11.9) | |
| Non-Hispanic Black | 1637 (22.0) | 176 (11.1) | 301 (14.8) | 422 (22.9) | 738 (37.4) | |
| Non-Hispanic White | 3502 (47.1) | 843 (53.4) | 1055 (51.7) | 856 (46.4) | 748 (37.9) | |
| Other | 1036 (13.9) | 227 (14.4) | 295 (14.5) | 262 (14.2) | 252 (12.8) | |
| Marriage status, n (%) | | | | | | < 0.001 |
| Marriage or living with partner | 4761 (64.0) | 1125 (71.2) | 1372 (67.3) | 1161 (62.9) | 1103 (55.9) | |
| Living alone | 2677 (36.0) | 455 (28.8) | 668 (32.7) | 685 (37.1) | 869 (44.1) | |
| Education, years, n (%) | | | | | | < 0.001 |
| <9 | 3915 (52.6) | 762 (48.2) | 1057 (51.8) | 992 (53.7) | 1104 (56) | |
| 9-12 | 2103 (28.3) | 472 (29.9) | 548 (26.9) | 505 (27.4) | 578 (29.3) | |
| >12 | 1420 (19.1) | 346 (21.9) | 435 (21.3) | 349 (18.9) | 290 (14.7) | |
| BMI, kg.m² | 33.3 ± 6.0 | 31.9 ± 4.9 | 32.7 ± 5.3 | 33.5 ± 6.0 | 34.9 ± 6.9 | < 0.001 |
| Smoking status, n (%) | | | | | | 0.05 |
| Never | 4062 (54.6) | 889 (56.3) | 1141 (55.9) | 999 (54.1) | 1033(52.4) | |
| Former | 2193 (29.5) | 458 (29) | 606 (29.7) | 535 (29) | 594 (30.1) | |
| Current | 1183 (15.9) | 233 (14.7) | 293 (14.4) | 312 (16.9) | 345 (17.5) | |
| Alcohol use, n (%) | | | | | | < 0.001 |
| Never | 1734 (23.3) | 341 (21.6) | 437 (21.4) | 429 (23.2) | 527 (26.7) | |
| Former | 1880 (25.3) | 332 (21) | 455 (22.3) | 487 (26.4) | 606 (30.7) | |
| Current | 3824 (51.4) | 907 (57.4) | 1148 (56.3) | 930 (50.4) | 839 (42.5) | |
| Diabetes, n (%) | 2104 (28.3) | 335 (21.2) | 473 (23.2) | 553 (30) | 743 (37.7) | < 0.001 |
| Hypertension, n (%) | 4127 (55.5) | 724 (45.8) | 1055 (51.7) | 1073 (58.1) | 1275 (64.7) | < 0.001 |
| CVD, n (%) | 1128 (15.2) | 132 (8.4) | 249 (12.2) | 318 (17.2) | 429 (21.8) | < 0.001 |
| Waist, cm | 110.3 ± 12.5 | 107.3 ± 11.0 | 109.1 ± 11.7 | 111.1 ± 12.5 | 113.3 ± 13.7 | < 0.001 |
| GGT, U/L | 25.0 (18.0, 37.0) | 26.0 (19.0, 40.0) | 25.0 (19.0, 37.0) | 25.0 (18.0, 37.0) | 23.0 (17.0, 35.0) | < 0.001 |
| Triglyceride, mg/dL | 165.0 (113.0, 241.0) | 185.0 (128.0, 279.0) | 173.0 (121.0, 246.0) | 163.0 (109.0, 236.0) | 147.0 (101.0, 214.0) | < 0.001 |
| Glucose, mg/dL | 117.1 ± 40.6 | 117.0 ± 45.7 | 115.0 ± 36.4 | 118.5 ± 43.8 | 118.1 ± 37.1 | 0.245 |
| HDL, mg/dL | 46.9 ± 12.8 | 45.2 ± 12.2 | 46.4 ± 12.6 | 47.3 ± 12.8 | 48.2 ± 13.2 | < 0.001 |
| Total Cholesterol, mg/dL | 199.8 ± 43.7 | 204.5 ± 41.8 | 201.6 ± 42.7 | 200.4 ± 45.9 | 193.7 ± 43.5 | < 0.001 |
| ALT, U/L | 23.0 (18.0, 32.0) | 26.0 (20.0, 37.0) | 25.0 (19.0, 33.0) | 23.0 (18.0, 30.0) | 20.0 (16.0, 27.0) | < 0.001 |
| Creatinine, μmol/L | 84.2 ± 44.8 | 78.8 ± 20.6 | 81.1 ± 24.8 | 83.3 ± 27.3 | 92.6 ± 76.2 | < 0.001 |

RDW: red blood cell distribution width; BMI: body mass index; CVD: cardiovascular disease.

RCS analysis showed a linear correlation between RDW and cardiovascular mortality (P for nonlinearity = 0.145) (Fig 2B).

Survival curve analysis further indicated a marked reduction in survival rates in the greater Q4 RDW group compared to the lowest group (P < 0.001) (Fig 3B).

The relationship between cardiovascular mortality and RDW was investigated using a subgroup analysis across various factors, including smoking status, age, BMI, sex, CVD, and

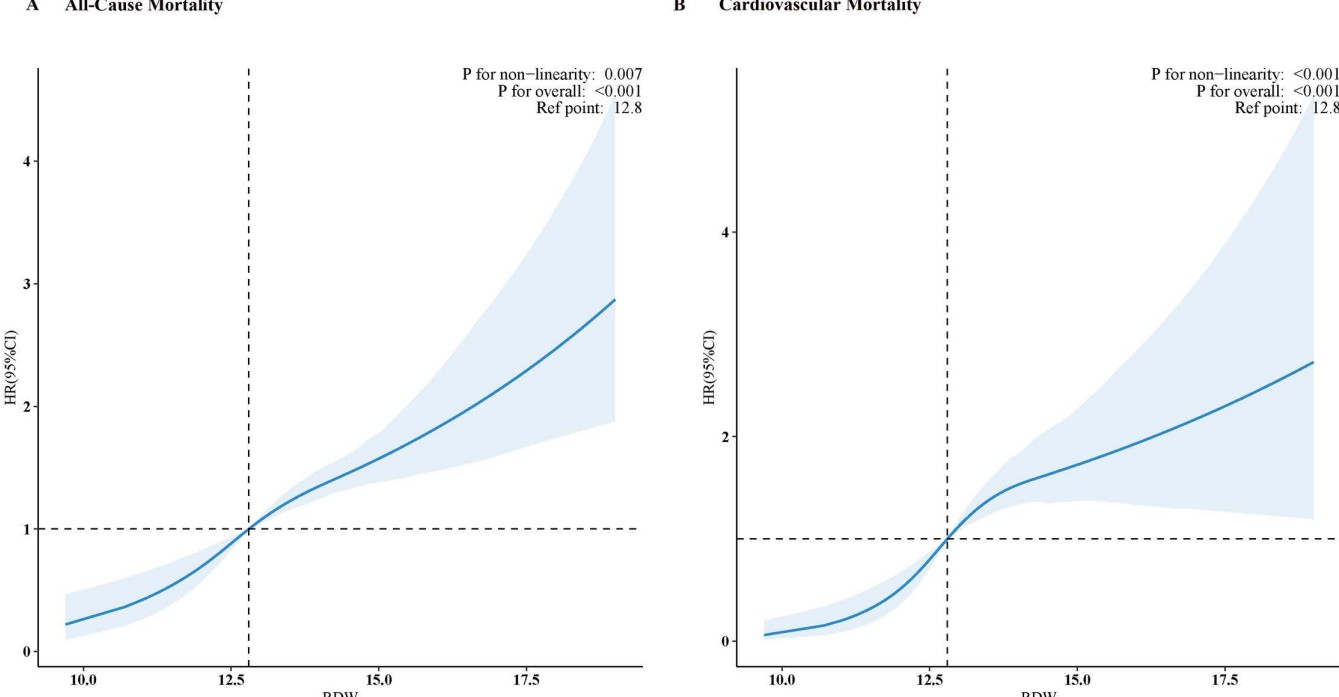

**Fig 2. The relationship between RDW and both all-cause** (A) and cardiovascular mortality (B) in individuals with NAFLD is depicted using restricted cubic splines. The hazard ratios were adjusted for various factors, including age, sex, race, BMI, smoking status, education level, diabetes, history of CVD.

diabetes. The primary findings were consistently observed across these subgroups, with no significant interactions detected (P > 0.05) (Fig 4B).

## The potential of RDW to forecast cardiovascular and all-cause death in individuals with NAFLD

Time-dependent ROC curve analysis showed that the AUC for RDW in predicting 3-, 5-, and 10-year all-cause mortality was f 0.69, 0.67, and 0.66, respectively (Fig 5A and 5B). The AUC for RDW in predicting 3-, 5-, and 10-year cardiovascular mortality was 0.70, 0.68, and 0.68, respectively (Figs 4C and 4D). These findings suggest that RDW has reliable prognostic potential for mortality throughout a range of time periods.

## Sensitivity analysis

Sensitivity analysis was conducted by including other serum markers (total cholesterol (TCHO), alanine aminotransferase (ALT), glucose (GLU), serum creatinine (SCr), and high-density lipoprotein (HDL-C)) in the multivariate analysis model. The results of weighted multivariate Cox regression analysis indicated that RDW was independently associated with all-cause and cardiovascular mortality (Table 3).

## Discussion

This study thoroughly examined the association between RDW and all-cause and cardiovascular mortality in a population with NAFLD using various methodologies. Analysis of data from 7,438 adult participants with NAFLD in the NHANES revealed a significant association between RDW and increased risk of all-cause and cardiovascular mortality.

**Table 2. The relationships between RDW and mortality in NALFD.**

| Characteristic | Model1 | | Model2 | | Model3 | |
|---|---|---|---|---|---|---|
| | HR(95% CI) | P value | HR(95% CI) | P value | HR(95% CI) | P value |
| **All-cause mortality** | | | | | | |
| RDW | 1.30 (1.24-1.36) | <0.001 | 1.24(1.16-1.33) | <0.001 | 1.22(1.14-1.31) | <0.001 |
| RDW quartile | | | | | | |
| Q1(≤12.2) | Reference | | Reference | | Reference | |
| Q2(12.3–12.7) | 1.75 (1.31-2.32) | <0.001 | 1.34 (1.01-1.79) | 0.046 | 1.35 (1.01-1.80) | 0.045 |
| Q3(12.8–13.4) | 2.93 (2.23-3.85) | <0.001 | 1.74 (1.35-2.24) | <0.001 | 1.59 (1.23-2.07) | <0.001 |
| Q4(≥13.5) | 4.87 (3.71-6.38) | <0.001 | 2.66 (1.99-3.56) | <0.001 | 2.29(1.72-3.06) | <0.001 |
| P for trend | <0.001 | | | <0.001 | | <0.001 |
| **Cardiovascular mortality** | | | | | | |
| RDW | 1.33 (1.27-1.39) | <0.001 | 1.27 (1.18-1.37) | <0.001 | 1.23 (1.14-1.33) | <0.001 |
| RDW quartile | | | | | | |
| Q1 (≤12.2) | Reference | | Reference | | Reference | |
| Q2 (12.3–12.7) | 2.59 (1.60-4.20) | <0.001 | 2.53 (1.52-4.21) | <0.001 | 1.95 (1.20-3.17) | 0.007 |
| Q3 (12.8–13.4) | 4.23 (2.59-6.91) | <0.001 | 4.39 (2.69-7.17) | <0.001 | 1.59 (1.21-3.61) | 0.009 |
| Q4 (≥13.5) | 8.26 (5.28-12.94) | <0.001 | 9.79 (6.20 -15.43) | <0.001 | 3.61(2.17-6.02) | <0.001 |
| P for trend | <0.001 | | | <0.001 | | <0.001 |

Model 1: unadjusted.

Model 2: adjusted for sex, age, race.

Model 3: adjusted for sex, age, race, marriage, education, smoking status, BMI, CVD, diabetes, hypertension.

RDW: red blood cell distribution width; HR: hazard ratio.

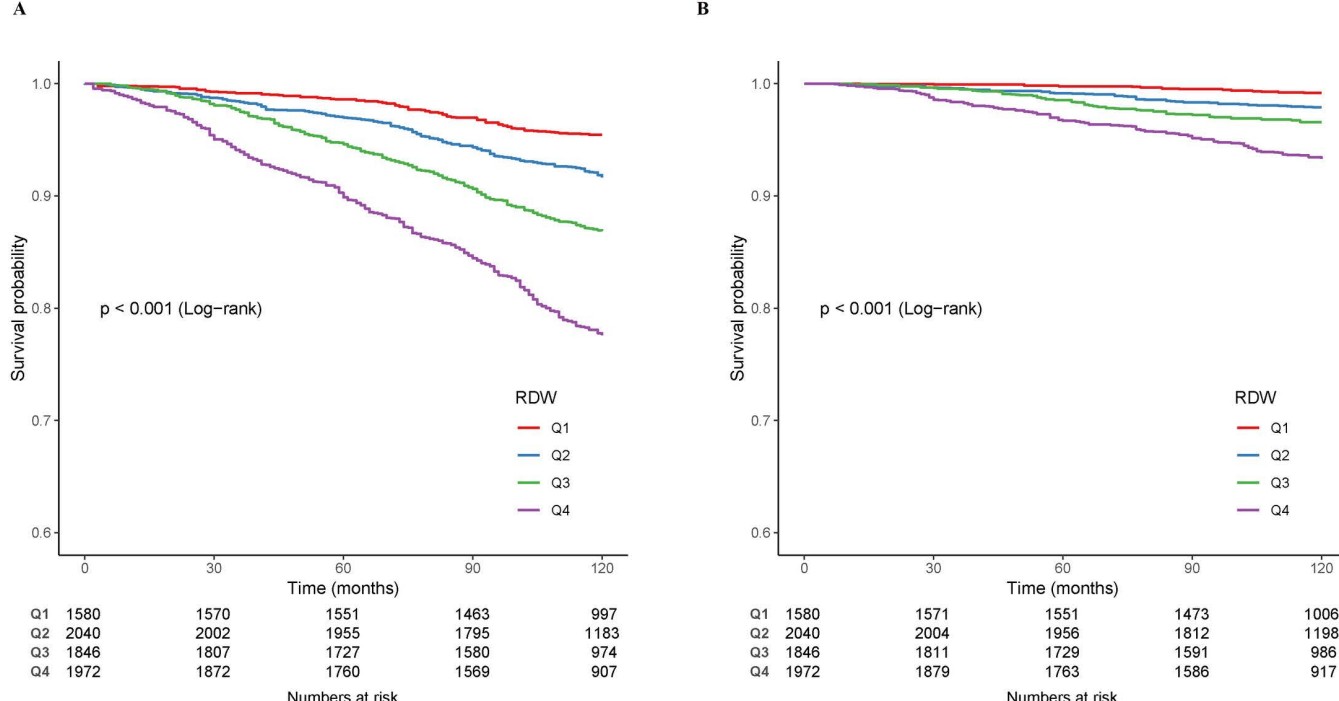

**Fig 3. Kaplan–Meier survival curves stratified by RDW quartiles, depicting survival rates for all-cause mortality** (A) and cardiovascular mortality (B).

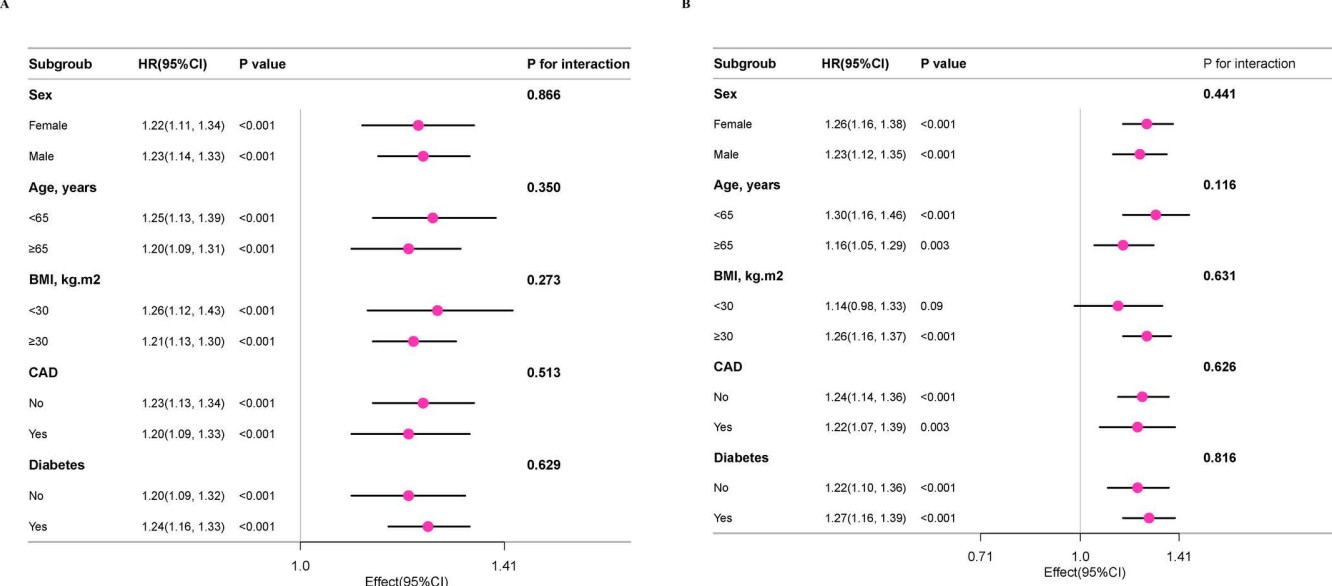

**Fig 4. Subgroup analysis across various factors of the association between RDW with all-cause mortality** (A) and cardiovascular mortality (B). subgroup analysis across various factors, including smoking status, age, BMI, sex, CVD, and diabetes.

RDW is a measure of the size distribution of red blood cell in circulation. An increase in RDW can be caused by any physiological mechanism that alters red blood cell shape and releases immature cells into circulation early. Previous studies have demonstrated that RDW can serve as a prognostic indicator of various diseases, including acute myocardial infarction [8], COVID-19 [9], heart failure [10,22], sepsis [11], hepatocellular carcinoma [23], acute respiratory failure [24], acute pancreatitis [25]. RDW has also been linked to all- cause mortality and specific mortality in the general population [26–28]. Consistent with these findings, this study demonstrated a positive association between RDW and all-cause mortality in individuals with NAFLD.

Previous research has demonstrated that elevated RDW is associated with cardiovascular mortality [29]. Katamreddy *et al.*[30] reported that an elevated RDW was associated with cardiovascular mortality in the intermediate ASCVD group in a study of 8884 subjects from NHANES III. Liao *et al.* [31] found that RDW was an independent predictor and showed a linear association with 1-year cardiovascular mortality in patients undergoing percutaneous coronary intervention (PCI). Lin *et al.* [32] identified RDW as an independent predictor of cardiovascular mortality in 181 STEMI patients (OR = 1.288, 95% CI, 1.126–1.472; P = 0.0005). This studys findings indicate that RDW is associated with cardiovascular mortality in the NAFLD population, aligning with the results reported above.

However, the reason for the association between increased RDW and poor outcomes in patients with NALFD is not been fully understood. Based on the literature, we propose the following potential mechanisms: A plausible pathway is the stimulation of proinflammatory cytokines, which are associated with NALFD [19]. An increase in RDW might result from the inhibition of erythropoietin-driven erythrocyte maturation caused by the rise of interleukin-1β, TNFα, and interleukin 6 [33]. In addition, increased RDW reflects a range of underlying metabolic problems, including inflammation and malnutrition, which can lead to decreased erythropoiesis and aberrant red blood cell survival [34]. Consequently, we postulated that RDW may represent inflammation, malnourishment, and other lifetime anomalies

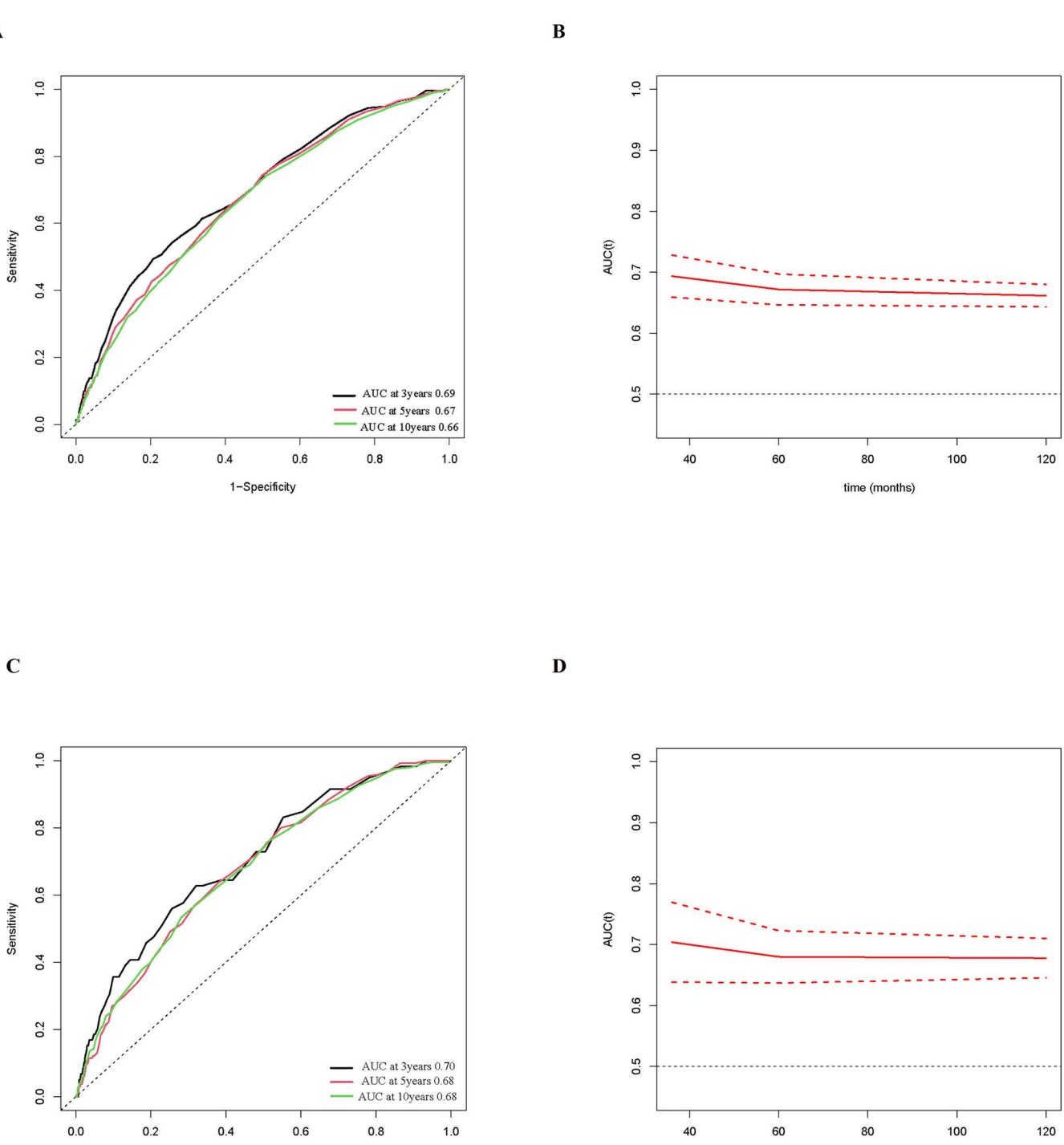

**Fig 5. Time-dependent ROC curves and corresponding AUC values, along with 95% confidence intervals, for RDW in predicting all-cause mortality** (A, B) and cardiovascular mortality (C, D).

**Table 3.** The relationships between RDW and mortality in NALFD, additionally adjusted for TCHO, ALT, GLU, SCr, and HDL-C.

| Characteristic | Crude model | | Adjusted model | |
|---|---|---|---|---|
| | HR(95% CI) | P value | HR(95% CI) | P value |
| RDW | 1.30 (1.24-1.36) | <0.001 | 1.21(1.13-1.30) | <0.001 |
| RDW quartile | | | | |
| Q1(≤12.2) | Reference | | Reference | |
| Q2(12.3–12.7) | 1.73 (1.30-2.31) | <0.001 | 1.38 (1.03-1.85) | 0.032 |
| Q3(12.8–13.4) | 2.92 (2.22-3.85) | <0.001 | 1.58 (1.22-2.04) | <0.001 |
| Q4(≥13.5) | 4.87 (3.68-6.35) | <0.001 | 2.20(1.64-2.95) | <0.001 |
| P for trend | | <0.001 | | <0.001 |
| RDW | 1.33 (1.27-1.39) | <0.001 | 1.21 (1.12-1.31) | <0.001 |
| Q1 (≤12.2) | Reference | | Reference | |
| Q2 (12.3–12.7) | 2.60 (1.60-4.24) | <0.001 | 2.06 (1.24-3.43) | 0.005 |
| Q3 (12.8–13.4) | 4.32 (2.63-7.08) | <0.001 | 2.19 (1.31-3.66) | 0.003 |
| Q4 (≥13.5) | 8.36 (5.32-13.13) | <0.001 | 3.44(2.06-5.75) | <0.001 |
| P for trend | | <0.001 | | <0.001 |

Adjusted model: adjusted for sex, age, race, marriage, education, smoking status, BMI, CVD, diabetes, hypertension, TCHO, ALT, GLU, SCr, and HDL-C.

RDW: red blood cell distribution width; HR: hazard ratio; TCHO: total cholesterol; ALT: alanine aminotransferase; GLU: glucose; SCr: serum creatinine; HDL-C: high-density lipoprotein.

may be linked to both cause-specific and all-cause mortality [7]. Further research is needed to fully elucidate the underlying biological mechanisms linking between higher RDW to adverse outcomes.

This study had several strengths. First, it is the first large-scale study to evaluate the relationship between RDW and mortality in adults with NAFLD living in the United States. Second, the research categorized RDW into distinct variables, which helped mitigate confounding factors and strengthened the reliability of the findings. However, this study also has limitations. First, some potential confounding factors that might influence the association between RDW and mortality may have been omitted from the analysis. Second, because the study data were from participants in the US, further research is needed to confirm whether the results can be generalized to populations with NALFD in other countries. Third, as judging RDW based purely on a single laboratory test may not be indicative of patient immunity, more clinical randomized controlled studies are required to verify our findings.

## Conclusion

In patients with NAFLD in the US, RDW is an independent predictor of increased risk of cardiovascular and all-cause mortality. Importantly, RDW demonstrated good predictive ability for both short-term and long-term mortality. Further prospective studies are required to confirm these findings and elucidate the underlying mechanisms.

## Acknowledgments

The authors sincerely thank the Physician Scientist Team for their enthusiastic and meticulous teaching and guidance on NHANSES study.

## Author contributions

**Conceptualization:** Yingxiu Huang, Ming Hu.

**Data curation:** Yingxiu Huang, Ting Ao, Yinying Wang, Peng Zhen.

**Formal analysis:** Yingxiu Huang, Ting Ao, Yinying Wang, Peng Zhen.

**Supervision:** Ming Hu.

**Writing – original draft:** Yingxiu Huang, Ting Ao, Yinying Wang, Peng Zhen.

**Writing – review & editing:** Ming Hu.

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
