## [Decision Letter · Decision Letter 0]

5 Jan 2025

PONE-D-24-46128The red blood cell distribution width is associated with all-cause and cardiovascular mortality among individuals with non-alcoholic fatty liver diseasePLOS ONE

Dear Dr. Hu, Thank you for submitting your manuscript to PLOS ONE. After careful consideration, we feel that it has merit but does not fully meet PLOS ONE’s publication criteria as it currently stands. Therefore, we invite you to submit a revised version of the manuscript that addresses the points raised during the review process.

**ACADEMIC EDITOR: ** We have received the reports from our advisors on your manuscript, The red blood cell distribution width is associated with all-cause and cardiovascular mortality among individuals with non-alcoholic fatty liver disease (PONE-D-24-46128). Based on the advice received, I feel that your manuscript could be reconsidered for publication should you be prepared to incorporate major revisions. When preparing your revised manuscript, you are asked to carefully consider the reviewer comments which can be found below, and submit a list of responses to the comments. You are kindly requested to also check the website for possible reviewer attachment(s). Please check online for eventual reviewer attachments.

We look forward to receiving your revised manuscript.

Kind regards,

Kovuri Umadevi

Academic Editor

PLOS ONE

Journal Requirements:

https://www.frontiersin.org/journals/nutrition/articles/10.3389/fnut.2024.1467824/full

In your revision ensure you cite all your sources (including your own works), and quote or rephrase any duplicated text outside the methods section. Further consideration is dependent on these concerns being addressed.

Additional Editor Comments:

Dear Ming Hu

We have received the reports from our advisors on your manuscript, The red blood cell distribution width is associated with all-cause and cardiovascular mortality among individuals with non-alcoholic fatty liver disease (PONE-D-24-46128).

Based on the advice received, I feel that your manuscript could be reconsidered for publication should you be prepared to incorporate major revisions.

When preparing your revised manuscript, you are asked to carefully consider the reviewer comments which can be found below, and submit a list of responses to the comments. You are kindly requested to also check the website for possible reviewer attachment(s). Please check online for eventual reviewer attachments.

Reviewers' comments:

Reviewer's Responses to Questions

**Comments to the Author**

1. Is the manuscript technically sound, and do the data support the conclusions?

Reviewer #1: Yes

2. Has the statistical analysis been performed appropriately and rigorously? 

Reviewer #1: Yes

3. Have the authors made all data underlying the findings in their manuscript fully available?

Reviewer #1: Yes

4. Is the manuscript presented in an intelligible fashion and written in standard English?

Reviewer #1: No

5. Review Comments to the Author

Reviewer #1: The authors used public data from the NHANES database to investigate the relationship between red blood cell distribution width (RDW) and all-cause and cardiovascular mortality in patients with non-alcoholic fatty liver disease (NAFLD).

1.The entire manuscript requires editing by a native English speaker for clarity and fluency.

2.The logical structure of the Introduction is not sufficiently clear, which may be due to language issues. For instance, the second and third paragraphs redundantly introduce RDW. A review of published literature indicates that there are numerous studies focused on the relationship between RDW and NAFLD, contradicting the authors’ claim that “There were few researches explored the relationship between RDW on NAFLD.”

3.In the Study Design and Participants section, the authors define excessive alcohol consumption as more than 4 cups per day for men and more than 3 cups for women. What is the basis for these definitions? What type of alcoholic beverages are included? These standards are crucial for accurately including patients with non-alcoholic cirrhosis.

4.Serum levels of CRP, TCHO, ALT, GLU, TG, SCr, and HDL-C should be included in the analysis model. Additionally, since the authors propose that RDW is an important inflammatory marker, it would be beneficial to provide correlation data between RDW and CRP.

5.The vertical axes in Figures 2A and 2B should be labeled as All-Cause Mortality and Cardiovascular Mortality, respectively.

6.The Discussion section is overly simplistic, particularly in its insufficient exploration of the relationship between RDW and cardiovascular mortality.

6. PLOS authors have the option to publish the peer review history of their article (what does this mean? ). If published, this will include your full peer review and any attached files.

**Do you want your identity to be public for this peer review?** For information about this choice, including consent withdrawal, please see our Privacy Policy .

Reviewer #1: No

---

## [Author Response · Author response to Decision Letter 0]

21 Jan 2025

Dear Editor and Reviewer:

We appreciate the opportunity to allow us to revise our manuscript and thanks for reviewers’ constructive comments and suggestions. We would like to submit our revised manuscript, entitled “The red blood cell distribution width is associated with all-cause and cardiovascular mortality among individuals with non-alcoholic fatty liver disease” (PONE-D-24-46128) for consideration for publication. In the revised manuscript, we have carefully addressed all comments and questions raised by reviewer’ point-by-point. We greatly appreciate your time and efforts to improve our manuscript for publication. 

Response to Editor

Response: Thank you very much for your suggestion. We have thoroughly reviewed the PLOS ONE style templates and have ensured that our manuscript adheres to all specified formatting requirements. We have double-checked the document to ensure consistency with the PLOS ONE guidelines.

https://www.frontiersin.org/journals/nutrition/articles/10.3389/fnut.2024.1467824/full

In your revision ensure you cite all your sources (including your own works), and quote or rephrase any duplicated text outside the methods section. Further consideration is dependent on these concerns being addressed.

Response: Thank you for your valuable feedback. We apologize for the overlap with our published article (PMID: 39421611) and have revised the Methods section to rephrase the content (page 8, line 138-151).

Evaluation of mortality from all causes and follow-up

Mortality status was determined by integrating information obtained from the National Death Index, which may be reached at https://www.cdc.gov/nchs/data-linkage/mortality-public.htm, with the NHANES data. Participants were classified as either alive or deceased based on NDI data. The follow-up period was calculated by subtracting the date of the NHANES examination from the date of death (December 31, 2019). Cardiovascular and all-cause mortality were retrieved and analyzed using the 10th Revision of the International Classification of Diseases (ICD-10). Cardiovascular mortality was defined as death due to cardiac conditions, which were categorized under the codes I00–I09, I11, I13, and I20–I51. The median follow-up period was 124 months (interquartile range: 100–147).

Response: Thank you for the detailed instructions provided. We have confirmed that the corresponding author (Ming Hu) now has an ORCID ID (0009-0002-4899-6751), which has been validated in Editorial Manager.

Response: Thank you for your guidance regarding the placement of the ethics statement in our manuscript. We have reviewed our manuscript and ensured that the ethics statement now appears only in the Methods section. Any mention of the ethics statement in other sections has been removed. 

Response to reviewer

1.The entire manuscript requires editing by a native English speaker for clarity and fluency.

Response: Thank you for your valuable suggestions. We fully agree that a native English speaker's input can significantly enhance the readability and coherence of our work. In response to your suggestion, we have engaged a professional native English editor with expertise in academic writing. This editor has carefully reviewed and revised the entire manuscript to ensure that the language is clear, concise, and fluent.

2. The logical structure of the Introduction is not sufficiently clear, which may be due to language issues. For instance, the second and third paragraphs redundantly introduce RDW. A review of published literature indicates that there are numerous studies focused on the relationship between RDW and NAFLD, contradicting the authors’ claim that “There were few researches explored the relationship between RDW on NAFLD.”

Response: We greatly appreciate the reviewer’s insightful comment. We acknowledge that the redundancy between the second and third paragraphs could lead to confusion, and we have revised this section to improve clarity and eliminate repetition (paragraph 2).

Red blood cell distribution width (RDW), a routinely measured parameter in a complete blood count (CBC), assesses the variation in the size of red blood cells (RBCs). RDW is calculated by determining the standard deviation of RBC volume relative to mean corpuscular volume (MCV). The normal range for RDW is typically 11–15%[7]. While low RDW values are generally not clinically significant, elevated RDW levels can be associated with increased levels of inflammatory markers such as erythrocyte sedimentation rate (ESR), C-reactive protein (CRP), and interleukins (IL)[7]. Numerous studies have demonstrated that RDW can serve as a prognostic marker for various diseases, including acute myocardial infarction[8], covid-19[9], heart failure[10], sepsis[11], and hepatocellular carcinoma[12]. Although several studies have explored the association between RDW and various conditions, fewer have focused on its relationship with NAFLD, Yang et al. [13]reported that patients with NAFLD tended to exhibit elevated RDW levels in a Chinese hospital cohort. However, in the United State, the relationship between RDW and mortality in patients with NAFLD remains unclear.

3.In the Study Design and Participants section, the authors define excessive alcohol consumption as more than 4 cups per day for men and more than 3 cups for women. What is the basis for these definitions? What type of alcoholic beverages are included? These standards are crucial for accurately including patients with non-alcoholic cirrhosis.

response: Thank you for your thoughtful comment. We understand the importance of clearly defining these parameters to ensure accurate inclusion of patients with non-alcoholic cirrhosis. And we revised the manuscript with clear definition of excessive alcohol intake.

(1).The definition of excessive alcohol consumption in our study is based on established references in the literature(16): excessive alcohol consumption as more than 4 drinks per day for men and more than 3 drinks for women. excessive alcohol intake[16] (≥4 drinks per day for men and ≥3 drinks per day for women; one drink is defined as a 12 ounce beer, a 5 ounce glass of wine, or 1.5 ounces of liquor, including whisky, gin, beer, wine, wine coolers, and any other alcoholic beverage (page 6, line 86-90)

(2). What type of alcoholic beverages are included?

Alcoholic beverages included are whiskey, gin, beer, wine, wine coolers, and any other alcoholic beverage (page 6, line 88-89).

4.Serum levels of CRP, TCHO, ALT, GLU, TG, SCr, and HDL-C should be included in the analysis model. Additionally, since the authors propose that RDW is an important inflammatory marker, it would be beneficial to provide correlation data between RDW and CRP.

(1) Response: We appreciate your thoughtful suggestions. Regarding the inclusion of CRP in the analysis model, we would like to clarify that CRP data is available only in the 2005-2010 NHANES cycles, and is missing in the 2011-2016 cycles. As a result, including CRP would lead to incomplete data for the cohort, and we were unable to incorporate it into our analysis model. While we acknowledge that CRP is an important inflammatory marker, the absence of data in the later cycles made it unfeasible to include it. Additionally, we chose not to include triglycerides (Tg) in the final model due to concerns about multicollinearity, as TG is closely associated with the definition of NAFLD. After careful consideration during model development, we excluded Tg to avoid potential issues arising from collinearity between predictors.

However, we have included other relevant serum markers, such as TCHO, ALT, GLU, SCr, and HDL-C, as part of a sensitivity analysis to provide a comprehensive understanding of the factors influencing RDW and its association with clinical outcomes. The sensitivity analysis results demonstrated that RDW was associated with both all-cause mortality and cardiovascular mortality (P<0.05) (Table 3).

Table 3 The relationships between RDW and mortality in NALFD, additionally adjusted for TCHO, ALT, GLU, SCr, and HDL-C

Characteristic Crude model Adjusted model

HR(95% CI) P value HR(95% CI) P value

All-cause mortality

RDW 1.30 (1.24-1.36) <0.001 1.21(1.13-1.30) <0.001

RDW quartile

Q1(≤12.2) Reference Reference

Q2(12.3-12.7)

1.73 (1.30-2.31) <0.001 1.38 (1.03-1.85) 0.032

Q3(12.8-13.4)

2.92 (2.22-3.85) <0.001 1.58 (1.22-2.04) <0.001

Q4(≥13.5)

4.87 (3.68-6.35) <0.001 2.20(1.64-2.95) <0.001

P for trend <0.001 <0.001

Cardiovascular mortality

RDW 1.33 (1.27-1.39) <0.001 1.21 (1.12-1.31) <0.001

Q1 (≤12.2) Reference Reference

Q2 (12.3-12.7)

2.60 (1.60-4.24) <0.001 2.06 (1.24-3.43) 0.005

Q3 (12.8-13.4)

4.32 (2.63-7.08) <0.001 2.19 (1.31-3.66) 0.003

Q4 (≥13.5)

8.36 (5.32-13.13) <0.001 3.44(2.06-5.75) <0.001

P for trend <0.001 <0.001

Adjusted model: adjusted for sex, age, race, marriage, education, smoke, BMI, CVD, diabetes, hypertension, TCHO, ALT, GLU, SCr, and HDL-C.

RDW: red blood cell distribution width; HR: hazard ratio.

(2) Additionally, since the authors propose that RDW is an important inflammatory marker, it would be beneficial to provide correlation data between RDW and CRP.

Response: Thank you for your valuable suggestion. We acknowledge that CRP is a well-established inflammatory marker, and examining its correlation with RDW could strengthen our argument regarding the inflammatory role of RDW in NAFLD.

To address this, we conducted a linear regression analysis assessing the association between RDW and CRP with data from 2005-2010 years. The results demonstrated RDW was positively associated with CRP (β 0.16, 95%CI: 0.12-0.21, P<0.001) (Table S1).

Table S1 The association between RDW and CRP

Variable N Crude model Adjusted model

Coefficient (95%CI) P value Coefficient (95%CI) P value

RDW 5740 0.14 (0.12~0.15) <0.001 0.16 (0.12~0.21) <0.001

Adjusted for age, sex, race, marital status, education, BMI, smoking status, alcohol use, CVD, diabetes, and hypertension.

5.The vertical axes in Figures 2A and 2B should be labeled as All-Cause Mortality and Cardiovascular Mortality, respectively.

Response: Thank you for pointing this out. We have revised Figures 2A and 2B to ensure that the vertical axes are correctly labeled as "A All-Cause Mortality" and "B Cardiovascular Mortality," respectively. 

6.The Discussion section is overly simplistic, particularly in its insufficient exploration of the relationship between RDW and cardiovascular mortality.

Response: Thank you for your insightful comment. We acknowledge that the relationship between RDW and cardiovascular mortality was not fully explored in the original manuscript. Based on your suggestion, we have revised the Discussion section to provide a more in-depth analysis of this relationship (page12-13, line230-239).

Previous research has demonstrated that elevated RDW is associated with cardiovascular mortality[28]. Katamreddy et al.[29] reported that an elevated RDW was associated with cardiovascular mortality in the intermediate ASCVD group in a study of 8884 subjects from NHANES III. Liao et al.[30] found that RDW was an independent predictor and showed a linear association with 1-year cardiovascular mortality in patients undergoing percutaneous coronary intervention (PCI). Lin et al.[31] identified RDW as an independent predictor of cardiovascular mortality in 181 STEMI patients (OR = 1.288, 95% CI, 1.126–1.472; P = 0.0005). This study’s findings indicate that RDW is associated with cardiovascular mortality in the NAFLD population, aligning with the results reported above.

Reference

16. Zhao E, Cheng Y, Yu C, Li H, Fan X. The systemic immune-inflammation index was non-linear associated with all-cause mortality in individuals with nonalcoholic fatty liver disease. Ann Med. 2023 Dec;55(1):2197652.

---

## [Decision Letter · Decision Letter 1]

31 Jan 2025

PONE-D-24-46128R1The red blood cell distribution width is associated with all-cause and cardiovascular mortality among individuals with non-alcoholic fatty liver diseasePLOS ONE

Dear Dr. Hu,

Thank you for submitting your manuscript to PLOS ONE. After careful consideration, we feel that it has merit but does not fully meet PLOS ONE’s publication criteria as it currently stands. Therefore, we invite you to submit a revised version of the manuscript that addresses the points raised during the review process.

Subject: Decision on Manuscript PONE-D-24-46128R1

Dear YUTA YOKOYAMA

We have received the reports from our advisors on your manuscript titled The Red Blood Cell Distribution Width is Associated with All-Cause and Cardiovascular Mortality Among Individuals with Non-Alcoholic Fatty Liver Disease ((Manuscript PONE-D-24-46128R1), which you submitted to PLOS ONE.)

Based on the advice received, the Editors feel that your manuscript could be reconsidered for publication, provided that minor revisions are incorporated.

When preparing your revised manuscript, please carefully address the reviewers’ comments, which are attached. Additionally, you are requested to submit a detailed response to each of the reviewers' comments. Please check online for any attached reviewer files.

Submission Guidelines for the Revised Manuscript:

Upload two identical versions of the revised manuscript:

One version should include all revisions highlighted in colored text for easy identification.

The other should be a clean version without highlights.

Submit your editable source files (e.g., Word, TeX).

Upload the response to the reviewers as a separate submission item under ‘Attachment to Manuscript.’

Please ensure that all required modifications are incorporated before resubmission.

Best regards,

Dr. Kovuri Umadevi

We look forward to receiving your revised manuscript.

Kind regards,

Kovuri Umadevi

Academic Editor

PLOS ONE

Journal Requirements:

Reviewers' comments:

Reviewer's Responses to Questions

**Comments to the Author**

1. If the authors have adequately addressed your comments raised in a previous round of review and you feel that this manuscript is now acceptable for publication, you may indicate that here to bypass the “Comments to the Author” section, enter your conflict of interest statement in the “Confidential to Editor” section, and submit your "Accept" recommendation.

Reviewer #2: All comments have been addressed

2. Is the manuscript technically sound, and do the data support the conclusions?

Reviewer #2: Yes

3. Has the statistical analysis been performed appropriately and rigorously? 

Reviewer #2: I Don't Know

4. Have the authors made all data underlying the findings in their manuscript fully available?

Reviewer #2: Yes

5. Is the manuscript presented in an intelligible fashion and written in standard English?

Reviewer #2: Yes

6. Review Comments to the Author

Reviewer #2: There are some very minor things to correct. covid-19 should be COVID-19,

The phrase 'marry statue' is presumably meant to be 'marriage status'

'smoke' in a footnote should be 'smoking'

7. PLOS authors have the option to publish the peer review history of their article (what does this mean? ). If published, this will include your full peer review and any attached files.

**Do you want your identity to be public for this peer review?** For information about this choice, including consent withdrawal, please see our Privacy Policy .

Reviewer #2: No

---

## [Author Response · Author response to Decision Letter 1]

13 Feb 2025

Dear Editor and Reviewer:

We are grateful for the opportunity to revise our manuscript and for the reviewers' valuable comments and suggestions. We would like to submit our revised manuscript, entitled 'The red blood cell distribution width is associated with all-cause and cardiovascular mortality among individuals with non-alcoholic fatty liver disease' (PONE-D-24-46128R1), for consideration for publication. In the revised version, we have addressed all the comments and questions raised by the reviewers. We greatly appreciate the time and effort invested by the editorial team and the reviewers to enhance the quality of our manuscript.

Response to Editor

Response: Thank you for your reminder. We have thoroughly reviewed all the references cited in our manuscript and have updated the 12th reference (see page 22, Reference [12]).

[12] G. Vidili, A. Zinellu, A.A. Mangoni, M. Arru, V. De Murtas, E. Cuccuru, A. Fancellu, P. Paliogiannis, Red cell distribution width as a predictor of survival in patients with hepatocellular carcinoma, Medicina (Kaunas) 60 (2024) 391. https://doi.org/10.3390/medicina60030391.

Response to reviewer

1. There are some very minor things to correct. covid-19 should be COVID-19,

The phrase 'marry statue' is presumably meant to be 'marriage status'

'smoke' in a footnote should be 'smoking'

Response: Thank you for your attention to detail. We have made the following corrections:

1. "covid-19" has been corrected to "COVID-19" (see page 5, line 63 and page 12, line 225).

2. The phrase "marry statue" has been changed to "marriage status" (see page 16, Table 1).

3. "smoke" in the footnote has been revised to "smoking status" (see page 16, Table 1, page 19, line 296 and 20, line 303).

---

## [Decision Letter · Decision Letter 2]

11 Mar 2025

The red blood cell distribution width is associated with all-cause and cardiovascular mortality among individuals with non-alcoholic fatty liver disease

PONE-D-24-46128R2

Dear Dr. Ming Hu

We’re pleased to inform you that your manuscript has been judged scientifically suitable for publication and will be formally accepted for publication once it meets all outstanding technical requirements.

Kind regards,

Kovuri Umadevi

Academic Editor

PLOS ONE

Additional Editor Comments (optional):

Dear Ming Hu,

We have received sufficient reviewer comments for the manuscript "The Red Blood Cell Distribution Width is Associated with All-Cause and Cardiovascular Mortality Among Individuals with Non-Alcoholic Fatty Liver Disease" (Manuscript Number: PONE-D-24-46128R2), you submitted to PLOS ONE. Based on the reviewers' feedback, we are pleased to inform you that the decision is Accept.

Congratulations on this outcome, and we appreciate your contribution to the journal.

Best regards,

Dr. Kovuri Umadevi

Reviewers' comments:

Reviewer's Responses to Questions

**Comments to the Author**

1. If the authors have adequately addressed your comments raised in a previous round of review and you feel that this manuscript is now acceptable for publication, you may indicate that here to bypass the “Comments to the Author” section, enter your conflict of interest statement in the “Confidential to Editor” section, and submit your "Accept" recommendation.

Reviewer #2: All comments have been addressed

2. Is the manuscript technically sound, and do the data support the conclusions?

Reviewer #2: Yes

3. Has the statistical analysis been performed appropriately and rigorously? 

Reviewer #2: I Don't Know

4. Have the authors made all data underlying the findings in their manuscript fully available?

Reviewer #2: Yes

5. Is the manuscript presented in an intelligible fashion and written in standard English?

Reviewer #2: Yes

6. Review Comments to the Author

Reviewer #2: RDW is not to do with cell shape but with cell size. You need to correct the statement in the discussion,

7. PLOS authors have the option to publish the peer review history of their article (what does this mean? ). If published, this will include your full peer review and any attached files.

**Do you want your identity to be public for this peer review?** For information about this choice, including consent withdrawal, please see our Privacy Policy .

Reviewer #2: No

---

## [Editor Report · Acceptance letter]

PONE-D-24-46128R2

PLOS ONE

Dear Dr. Hu,

I'm pleased to inform you that your manuscript has been deemed suitable for publication in PLOS ONE. Congratulations! Your manuscript is now being handed over to our production team.

Kind regards,

on behalf of

Dr. Kovuri Umadevi

Academic Editor

PLOS ONE